# Beneficial effects of benfotiamine, a NADPH oxidase inhibitor, in isoproterenol-induced myocardial infarction in rats

**Lamiaa A. Ahmed**[1]*, **Omnia F. Hassan**[2], **Omneya Galal**[3], **Dina F. Mansour**[4], **Aiman El-Khatib**[1]

**1** Department of Pharmacology and Toxicology, Faculty of Pharmacy, Cairo University, Giza, Egypt,
**2** Department of Pharmacology and Toxicology, Faculty of Pharmacy, MSA University, 6th of October City, Egypt, **3** Department of Pharmacology and Toxicology, Faculty of Pharmacy, Ahram Canadian University, 6th of October City, Egypt, **4** Department of Pharmacology, Medical Research Division, National Research Centre, Egypt

* lamiaa.ahmed@pharma.cu.edu.eg

**Data Availability Statement:** All relevant data are within the manuscript.

**Funding:** The authors received no specific funding for this work.

## Abstract

### Background

Acute myocardial infarction (AMI) remains the most common cause of morbidity and mortality worldwide. The present study was directed to investigate the beneficial effects of benfotiamine pre- and post-treatments in isoproterenol (ISO)-induced MI in rats.

### Methods

Myocardial heart damage was induced by subcutaneous injection of ISO (150 mg/kg) once daily for two consecutive days. Benfotiamine (100 mg/kg/day) was given orally for two weeks before or after ISO treatment.

### Results

ISO administration revealed significant changes in electrocardiographic recordings, elevation of levels of cardiac enzymes; creatinine kinase (CK-MB) and troponin-I (cTn-I), and perturbation of markers of oxidative stress; nicotinamide adenine dinucleotide phosphate (NADPH) oxidase, malondialdehyde (MDA), reduced glutathione (GSH), superoxide dismutase (SOD) and glutathione peroxidase (GPx) and markers of inflammation; protein kinase C (PKC), nuclear factor-kappa B (NF-κB) and metalloproteinase-9 (MMP-9). The apoptotic markers (caspase-8 and p53) were also significantly elevated in ISO groups in addition to histological alterations. Groups treated with benfotiamine pre- and post-ISO administration showed significantly decreased cardiac enzymes levels and improved oxidative stress, inflammatory and apoptotic markers compared to the ISO groups.

**Competing interests:** NO authors have competing interests.

## Conclusion

The current study highlights the potential role of benfotiamine as a promising agent for prophylactic and therapeutic interventions in myocardial damage in several cardiovascular disorders via NADPH oxidase inhibition.

## 1. Introduction

Acute myocardial infarction (AMI) remains the most leading cause of morbidity and mortality worldwide. Myocardial infarction (MI) is an acute condition of heart muscle necrosis that happens as a result of inadequate balance between coronary blood supply as well as cardiac demand, leading to myocardial ischaemic injury and damage to cardiomyocytes [1,2]. Following the ischaemic event, inflammation mediates further myocardial tissue damage [3] through neutrophil infiltration in the infarcted area, where cardiomyocyte damage is triggered via the release of proteolytic enzymes and reactive oxygen species (ROS) generation [4]. Nicotinamide adenine dinucleotide phosphate (NADPH) oxidase is an enzyme complex that is responsible for the generation of a considerable amount of ROS [5]. This enzyme plays an essential role in isoproterenol (ISO)-mediated ROS production and myocardial cytotoxicity; consequently, its inhibition could represent a promising therapeutic target for the treatment of myocardial damage [6].

ISO is a synthetic sympathomimetic catecholamine that acts as a non-selective β-adrenergic agonist. Catecholamines exert different effects according to the dose used. At low doses, it can be useful for the treatment of bradycardia, heart block and bronchial asthma [3]. At high doses, ISO causes an inadequate balance between the production of free radicals as well as the antioxidative defence system [7]. Furthermore, adrenochrome and hydroxyl radicals are oxidative products of catecholamines that are involved in the pathogenesis of myocardial ischaemia. Following ISO administration, a robust decrease in the activities of endogenous antioxidant systems of the heart leads to the gradual accumulation of oxidative damage in cardiomyocytes [8,9]. It has been proven that β-adrenoceptor stimulation provokes NADPH oxidase-derived ROS production in the heart [5]. The excessively produced concentrations of ROS are responsible for the stimulation of the damaging inflammatory as well as apoptotic pathways [8].

Oxidative stress induces inflammation through activation of the transcription factors, including nuclear factor-kappa B (NF-κB) as well as mitogen-activated protein kinase (MAPK) signalling, which is involved in the expression of NADPH oxidase which ultimately contributes to cardiac inflammation, remodelling and failure [10].

Benfotiamine (S-benzoyl thiamine O-monophosphate), an acyl derivative of thiamine, is a known inhibitor of NADPH oxidase and has been reported to prevent tissue damage in numerous experimental models [11,12]. Benfotiamine has been shown to offer protection against diabetes-related complications including neuropathy, nephropathy and retinopathy [11]. Benfotiamine has been confirmed not only to directly inhibit NADPH oxidase activity but also to prevent the pathway of protein kinase C (PKC), thus blocking the activation of NF-κB in patients with diabetes [13]. Additionally, the inhibitory effect of benfotiamine on NADPH oxidase can occur indirectly via the activation of transketolase enzyme that eventually inhibits the production of NADPH oxidase and activates the antioxidant defence mechanisms [11]. Therefore, the purpose of the current study was to evaluate the beneficial effects of benfotiamine, a NADPH oxidase inhibitor, as a pre- and post-treatment in ISO-induced MI in rats.

## 2. Materials and methods

### 2.1 Animals

Male adult Wistar rats weighing 150–200 g were obtained from the Egyptian Organization for Biological Products and Vaccines (Cairo, Egypt). Rats were kept in the animal house of Faculty of Pharmacy, MSA University throughout the study period. They were housed in plexiglass cages under a controlled temperature of 25°C (25 ± 2°C) and a constant (12/12 h light/dark) cycle condition in the animal room and were allowed free access to water as well as a standard pellet diet. Appropriate indicators of animal health and well being are regularly monitored and tested in accordance with guidelines provided by the Ethics Committee for Animal Experimentation at Faculty of Pharmacy, Cairo University. The investigation complied with the Guide for Care and Use of Laboratory Animals published by the US National Institutes of Health (NIH Publication No. 85–23, revised 2011) and was approved by the Ethics Committee for Animal Experimentation at Faculty of Pharmacy, Cairo University (Permit Number: PT 2096).

### 2.2 Drugs and chemicals

Isoproterenol-HCl and benfotiamine (thiamine derivative) were obtained from Sigma-Aldrich (MO, USA) with CAS Registry Numbers# 51-30-9 and 22457-89-2 respectively. All other used chemicals and reagents, unless otherwise specified, were obtained from Sigma-Aldrich Chemical Co. (St. Louis, MO, USA).

### 2.3 Experimental design

Forty male Wistar albino rats were randomly allocated into five groups 8 animals each. Group I served as normal animals; rats received the ISO vehicle (0.9% NaCl) for 14 days. Group 2 (ISO-A); rats were kept for 2 weeks and then were given ISO (150 mg/kg/day, s.c.) for 2 consecutive days. Group 3 (prophylactic group); animals were pre-treated with benfotiamine (100 mg/kg/day, p.o.) for 2 weeks before and during ISO administration. Group 4 (ISO-B); animals directly received ISO (150 mg/kg/day, s.c.) for 2 consecutive days and then were kept for 2 weeks. Group 5 (treated group); rats were post-treated with benfotiamine (100 mg/kg/day, p. o.) for 2 weeks [14] after the administration of ISO.

At the end of the experiment, electrocardiogram (ECG) was monitored, as well as blood samples were then collected. Hearts were isolated rapidly, dissected and washed immediately with ice-cold saline. One portion was homogenized in phosphate buffer (0.1 N Tris HCl buffer, pH 7.4) to prepare a 10% (w/v) homogenate that was used for the biochemical estimation, while the cardiac apex was fixed in 10% formalin for histopathological examination. The cardiac protein contents of tissue homogenate, was estimated by using Lowry et al. method [15].

### 2.4 Induction of MI

MI was induced by S.C. injection of isoproterenol hydrochloride dissolved in normal saline at a high dose of 150 mg/kg once daily for two consecutive days. The used dose and route of administration were chosen from the published literatures [1,16].

### 2.5 ECG monitoring

Rats were anaesthetized with thiopental (45 mg/kg, i.p.), then kept warm with a heating lamp to avoid hypothermia. Needles of peripheral limb electrodes were inserted subcutaneously into fore-paw pads of each rat, and connected to an electrocardiograph (HPM 7100, Fukuda

Denshi, Tokyo, Japan) with a built in recording software in order to assess heart rate (HR), ST elevation, QT interval and QRS duration.

## 2.6 Biochemical assays

Serum levels of creatinine kinase (CK-MB) and troponin-I (cTn-I) were determined by ELISA kits (Cusabio, China; Cat. # CSB-E14403r; CSB-E08594r, respectively). The kits employed Double Antibody Sandwich Technique. The kits procedures were performed in line with the manufacturer's instructions then expressed as ng/ml and pg/ml, respectively. The used assays were sensitive up to 0.078 ng/ml and 7.81 pg/ml, respectively.

The activity of NADPH oxidase enzyme in the cardiac tissue was measured by using the lucigenin chemiluminescence method [17] with a microplate photometer (Thermo Fisher Scientific, Vantaa, Finland) and expressed as RLU /min/mg protein.

The cardiac content of oxidative stress biomarkers, including reduced glutathione (GSH), was assessed colorimetrically by using commercially available standard kit (Eagle biosciences, INC, China). The procedures of the used kit were performed along with the manufacturer's instructions, then the results were expressed as μg/g wet/tissue. The cardiac content of malondialdehyde (MDA) was assessed using a commercial kit (Eagle biosciences, INC, China) where results were measured colorimetrically at 586 nm and expressed as nmol/mg protein. The used assay was sensitive up to 0.1 nmol/ml. The cardiac activity of superoxide dismutase (SOD) was assessed by using a commercially standard colorimetric activity kit (Biodiagnostic, Egypt). The procedures of the used kit were performed in line with the manufacturer's instructions and the results were expressed as U/mg protein while the activity of glutathione peroxidase (GPx) was determined by using a commercially available standard kit (Bioassay systems, USA). The procedure was performed along with the manufacturer's instructions and the results were expressed as U/mg protein where the sensitivity using this assay was up to 40 U/l.

The inflammatory markers PKC, NF-κB and metalloproteinase -9- (MMP-9) were assessed by using commercially standard ELISA kits (Cusabio, China; Cat. # CSB-E12801r; MBS453975; CSB-E08008r, respectively). The procedures of the used kits were performed along with the manufacturer's instructions and the results were expressed as pg/mg protein, ng/mg protein as well as pg/mg protein, respectively. The used assays were sensitive up to 3.9 pg/ml, 0.119 ng/ml and 3.9 pg/ml, respectively.

Apoptotic markers; p53 and caspase-8 were assessed by using commercially standard ELISA kits (Cusabio, China; Cat. # CSB-E08336r; CSB-E14912r, respectively) according to the manufacturer's instructions and expressed as pg/mg of protein and ng/mg protein, respectively. The used assays were sensitive up to 3.12 pg/ml and 0.078 ng/ml, respectively.

## 2.7 Histopathologic assessment of myocardial damage

At the end of the experiment, six rats of each group were used for the histopathological examination. Rats were anaesthetized with thiopental (45 mg/kg, i.p.) followed by cervical dislocation. Samples were prepared from the hearts of all groups and then fixed in 10% formalin prepared in saline. Heart specimens were washed in saline, dehydrated in ascending grades of alcohol (ethanol), cleared in xylene then finally embedded in paraffin for 24 h. Sections of 3–5 μm were cut by a slide microtome, subjected to double staining with haematoxylin and eosin (H&E) to study and examine the general structure of the heart microscopically (magnification x200) using an image analyser (Leica Qwin 550, Germany). Myocardial tissue damage was assessed using a semi-quantitative scoring scale of 0–5 [18].

## 2.8 Statistical analysis

All data were assessed for normality as well as homogeneity of variance using Kolmogorov-Smirnov and Bartlett's tests, respectively. The results are expressed as the mean of 8 experiments ± SEM. Statistical significance was assessed using One-way analysis of variance (ANOVA) followed by Tukey's post hoc test, except for the histological score of tissue damage, which was expressed as median (interquartile changes) and was assessed using non-parametric one-way ANOVA (Kruskal-Wallis test) followed by multiple comparison Dunn's test. Statistical analysis was done using GraphPad Prism software (version 6.04). The level of significance was fixed at $p < 0.05$, for all statistical tests.

# 3. Results

## 3.1 Effect of benfotiamine pre- and post-treatments on heart weight index and ECG changes in ISO-induced MI in rats

Both ISO-A and ISO-B groups revealed a significant rise ($P<0.05$) in heart weight index (HWI), indicating hypertrophy that was significantly improved by benfotiamine pre- and post-treatments. Correspondingly, ISO-A group revealed a significant increase ($P<0.05$) in HR, ST elevation and T wave inversion along with significant prolongation of QTc interval in addition to QRS duration, indicating ischaemia as well as conduction abnormalities. Similarly, ISO-B group exerted significant changes in ECG parameters similar to ISO-A group, except for depression in ST segment amplitude, which is indicative of the progression of ischaemia-induced MI. On the other hand, benfotiamine pre- and post-treatments succeeded in improving ECG abnormalities secondary to ISO administration (Table 1).

## 3.2 Biochemical parameters

**3.2.1 Effect of benfotiamine pre- and post-treatments on cardiac enzyme markers in ISO-induced MI in rats.** Administration of ISO for two consecutive days (designated as ISO-A group) produced approximately four-fold increase in the serum cTn-I level, while the other group designated as ISO-B group showed a significant increase ($P<0.05$) in its level ($238.00 \pm 17.42$ versus $100.00 \pm 10.41$ pg/ml) in comparison to the normal group. Pre-and post-treatments with oral benfotiamine (100 mg/kg/day) normalized the cTn-I level (Fig 1A).

**Table 1. Effect of benfotiamine pre- and post-treatments on HWI and ECG changes in ISO-induced MI in rats.**

| Treatment groups | Normal | ISO-A | ISO-A+Benf. | ISO-B | ISO-B+Benf. |
|---|---|---|---|---|---|
| HWI (mg/g) | 3.28 ± 0.03 | 4.74* ± 0.23 | 3.85@ ± 0.15 | 4.62* ± 0.18 | 2.98# ± 0.14 |
| Heart rate (bpm) | 339.6 ± 3.9 | 486.5* ± 8.4 | 393.3*@ ± 8.3 | 410.3* ± 7.7 | 382.2* ± 5.5 |
| ST-height amplitude (mV) | 0.034 ± 0.007 | 0.093*±0.009 | 0.075 ± 0.005 | -0.13*± 0.008 | 0.027 #± 0.002 |
| T wave amplitude (mV) | 0.103±0.01 | -0.025*±0.012 | 0.028*@±0.004 | -0.015*±0.002 | 0.013*±0.001 |
| QTc (ms) | 120.9 ± 4.4 | 199.9* ±11.6 | 120.9@± 8.9 | 261.8* ± 4.5 | 123.4#± 5.4 |
| QRS duration (ms) | 22.2 ±1.7 | 47.72* ± 2.4 | 22.5@ ± 1.2 | 28.37 ± 1.9 | 19.50#± 0.9 |

Each value represents the mean of 8 experiments ± SEM (n = 8). Statistical analysis was done using One-way ANOVA followed by Tukey's post-hoc test.

*p<0.05 vs. normal group

@p<0.05 vs. ISO-A

#p<0.05 vs. ISO-B group. ISO-A: ISO pre-treatment control; ISO-B: ISO post-treatment control; ISO-A+ Benf.: benfotiamine pre-treatment; ISO-A+ Benf.: benfotiamine post-treatment.

On the other hand, ISO-A group showed approximately two fold increase ($P<0.05$) in serum CK-MB level, while ISO-B group did not show a significant change in serum CK level. The benfotiamine pre-treatment group completely alleviated the serum CK-MB level (Fig 1B).

**3.2.2 Effect of benfotiamine pre- and post-treatments on myocardial NADPH oxidase and oxidative stress markers in ISO-induced MI in rats.** ISO-A and ISO-B groups revealed significant increase ($P<0.05$) in myocardial NADPH oxidase activity ($11.87 \pm 0.57$ versus $4.50 \pm 0.11$ RLU/min/mg protein) and ($8.33 \pm 0.30$ versus $4.50 \pm 0.11$ RLU/min/mg protein), respectively. Benfotiamine pre- and post-treatments showed normalization and complete protection against such an increase (Fig 2A).

Administration of ISO in groups ISO-A and ISO-B significantly increased ($P<0.05$) cardiac MDA content ($26.12 \pm 2.52$ versus $1.97 \pm 0.09$ nmol/mg protein) and ($10.03 \pm 1.12$ versus $1.97 \pm 0.09$ nmol/mg protein), respectively compared to the normal group. In contrast, ISO-A and ISO-B groups showed a significant reduction ($P<0.05$) of myocardial GSH content

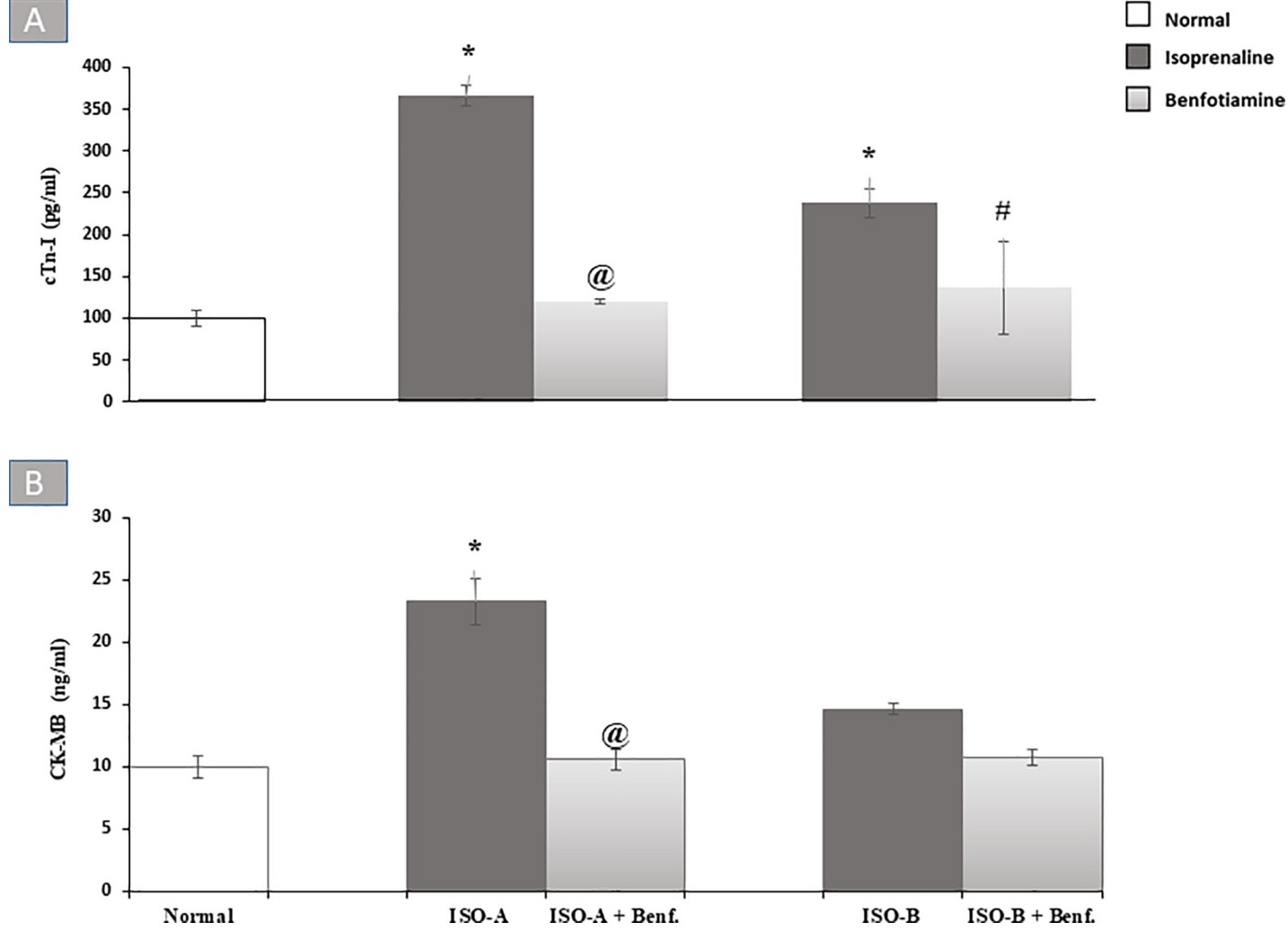

**Fig 1.** Effect of benfotiamine pre- and post-treatments on serum (A) cTn-I and (B) CK-MB levels in ISO-induced MI in rats. Each bar represents mean ± SEM (n = 8). Statistical analysis was done using One-way ANOVA followed by Tukey's post-hoc test. *significantly different from the normal group at p< 0.05. @significantly different from the ISO-A group at p< 0.05. #significantly different from the ISO-B group at p< 0.05. ISO-A: ISO pre-treatment control; ISO-B: ISO post-treatment control; ISO-A + Benf.: benfotiamine prophylactic; ISO-A+ Benf.: benfotiamine treatment.

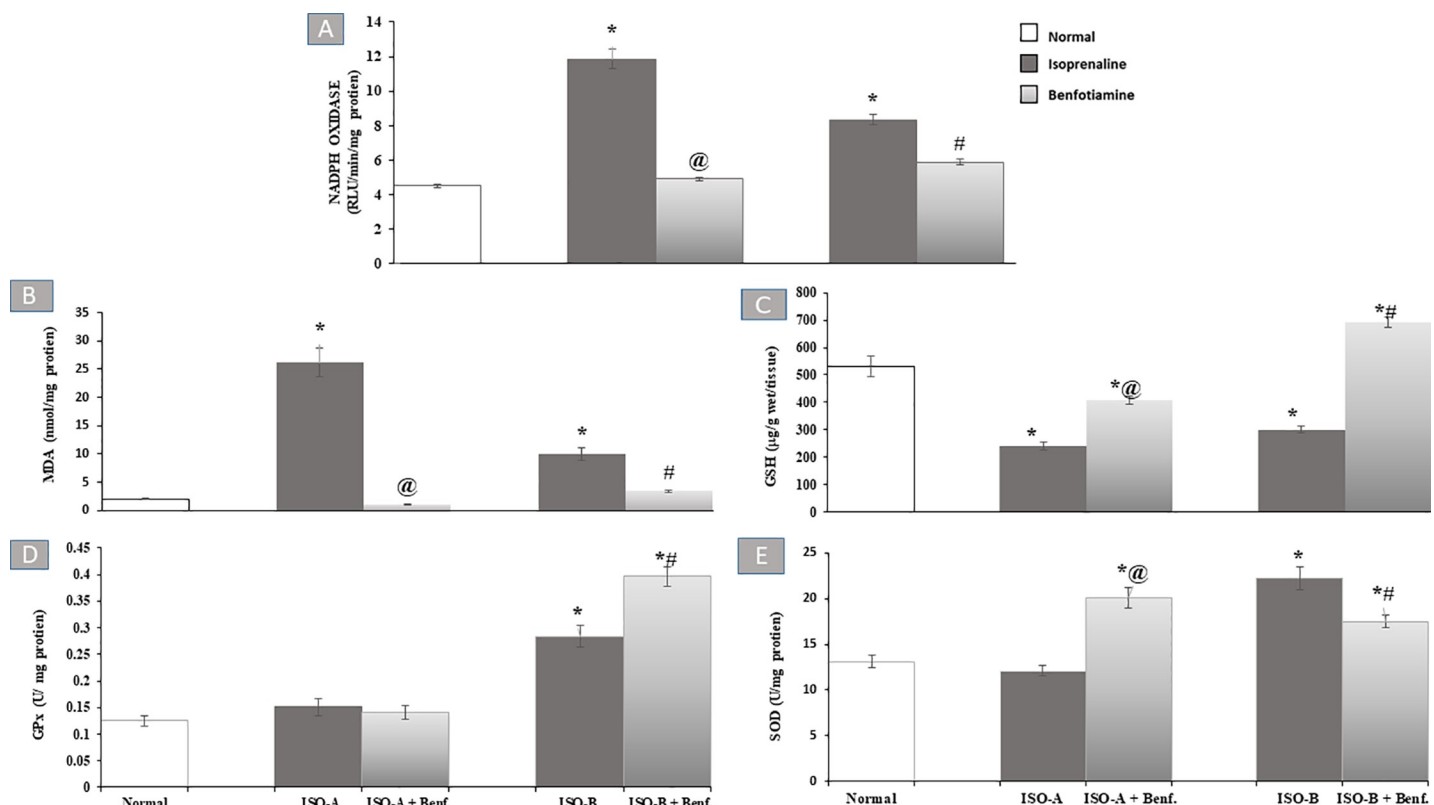

**Fig 2.** Effect of benfotiamine pre- and post-treatments on myocardial (A) NADPH oxidase activity, (B) MDA, (C) GSH contents, in addition to (D) GPx as well as (E) SOD activity in ISO-induced MI in rats. Each bar represents mean ±SEM (n = 8). Statistical analysis was done using One-way ANOVA followed by Tukey's post-hoc test. *significantly different from the normal group at p< 0.05. @significantly different from the ISO-A group at p< 0.05. #significantly different from the ISO-B group at p< 0.05. ISO-A: ISO pre-treatment control; ISO-B: ISO post-treatment control; ISO-A+ Benf.: benfotiamine prophylactic; ISO-A+ Benf.: benfotiamine treatment.

(240.97 ± 15.77 versus 532.76 ± 37.83 μg/g wet/tissue) and (300.69 ± 13.77 versus 532.76 ± 37.83 μg/g wet/tissue), respectively, compared to the normal group. Furthermore, ISO-A group showed a non-significant change in GPx and SOD activities whereas ISO-B group revealed a significant increase ($P<0.05$) in the myocardial activity of GPx and SOD (0.28 ± 0.02 versus 0.13 ± 0.01 U/mg protein) and (22.23 ± 1.20 versus 13.09 ± 0.69 U/mg protein), respectively, compared to the normal group. Prophylactic and therapeutic treatment with benfotiamine succeeded to normalize tissue content of MDA. Similarly, benfotiamine post-treatment completely improved perturbations of GSH content as well as GPx and SOD activities with further significant increase ($P<0.05$) in their values compared to the normal group. Meanwhile, benfotiamine pre-treatment group revealed a significant increase ($P<0.05$) in myocardial GSH content and a further significant elevation ($P<0.05$) of myocardial SOD activity compared to the normal group (Fig 2B–2E).

**3.2.3 Effect of benfotiamine pre- and post-treatments on myocardial contents of inflammatory markers in ISO-induced MI in rats.** Regarding inflammatory markers, ISO-A and ISO-B groups revealed a significant elevation (P<0.05) of the myocardial contents of PKC and MMP-9 compared to the normal group. Similarly, ISO-A group demonstrated approximately eightfold increase in myocardial NF-κB content. ISO-B group also showed a marked increase (P<0.05) in the myocardial NF-κB content (93.93 ± 3.82 versus 29.72 ± 2.13 ng/mg protein) compared to the normal group. Pre- and post-treatments with benfotiamine revealed marked protection and normalization of PKC and NF-κB contents. Likewise, the benfotiamine pre-

treatment revealed complete protection and normalization of MMP-9 content, while the post-treated group showed only a significant reduction (P<0.05) of its content compared to ISO-B group (Fig 3).

**3.2.4 Effect of benfotiamine pre- and post-treatments on apoptotic markers in ISO-induced MI in rats.** ISO-A group showed an approximately six-fold increase in myocardial p53 content and ISO-B group significantly increased (P<0.05) the myocardial p53 content (163.10 ± 11.59 versus 74.28 ± 3.61 pg/mg protein) compared to the normal group. The

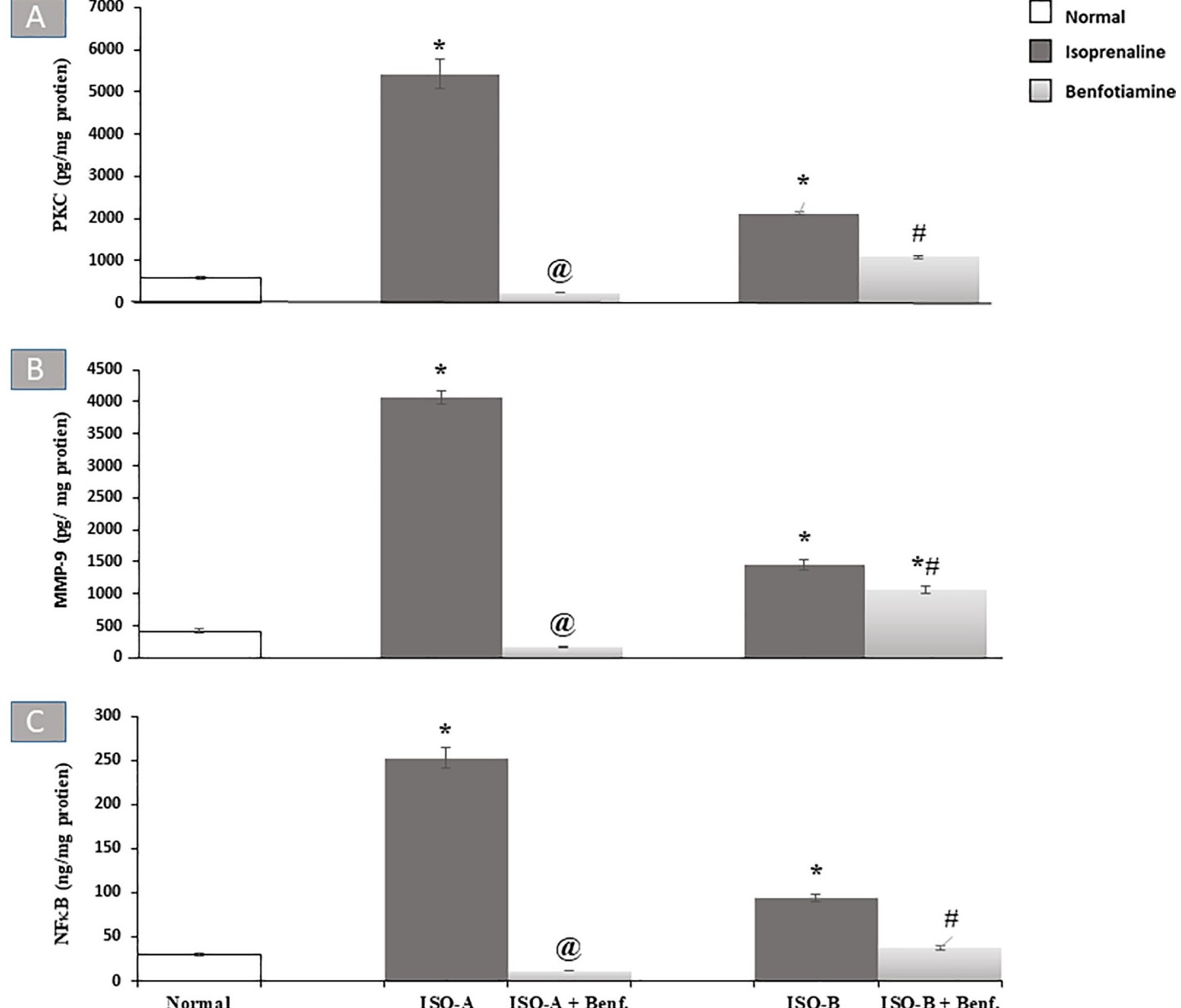

**Fig 3.** Effect of oral benfotiamine pre- and post-treatments on myocardial (A) PKC, (B) MMP-9 and (C) NF-$\kappa$B contents in ISO-induced MI in rats. Each bar represents mean ± SEM (n = 8). Statistical analysis was done using One-way ANOVA followed by Tukey's post-hoc test. *significantly different from the normal group at p< 0.05. @significantly different from the ISO-A group at p< 0.05. #significantly different from the ISO-B group at p< 0.05. ISO-A: ISO pre-treatment control; ISO-B: ISO post-treatment control; ISO-A+ Benf.: benfotiamine prophylactic; ISO-A+ Benf.: benfotiamine treatment.

benfotiamine pre-treated group revealed complete protection and normalization of p53 and caspase-8 contents. In contrast, the benfotiamine post-treated group showed no significant change in p53 content in comparison to ISO-B group (Fig 4A) whereas it completely ameliorated perturbation in myocardial caspase-8 content (Fig 4B).

### 3.3 Effect of benfotiamine pre- and post-treatments on histological changes in ISO-induced MI in rats

ISO groups showed marked cardiac muscle damage together with loss of nuclei as well as areas of infarction and pyknotic nuclei with cardiac lesion scores of 5 and 4 for ISO-A and ISO-B treated groups, respectively. Benfotiamine pre-treatment revealed nearly normalized cardiac architecture. Similarly, the benfotiamine post-treatment group improved the histopathological picture compared to ISO-B group, with a cardiac lesion score of 1 (Fig 5).

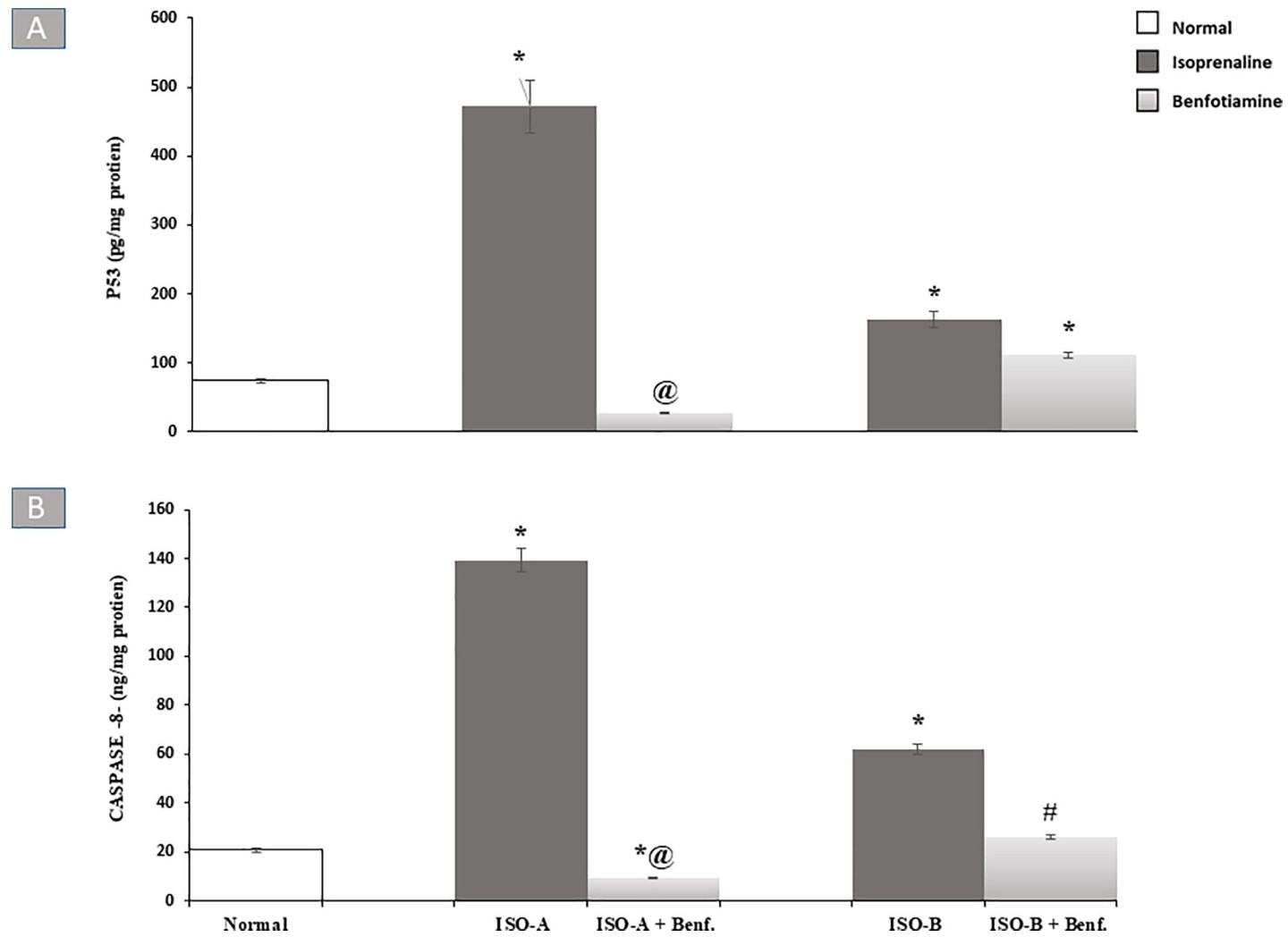

**Fig 4.** Effect of benfotiamine pre- and post-treatments on myocardial (A) p53 and (B) caspase-8- contents in ISO-induced MI in rats. Each bar represents mean ± SEM (n = 8). Statistical analysis was done using One-way ANOVA followed by Tukey's post-hoc test. *significantly different from the normal group at p< 0.05. @significantly different from the ISO-A group at p< 0.05. #significantly different from the ISO-B group at p< 0.05. ISO-A: ISO pre-treatment control; ISO-B: ISO post-treatment control; ISO-A+ Benf.: benfotiamine prophylactic; ISO-A+ Benf.: benfotiamine treatment.

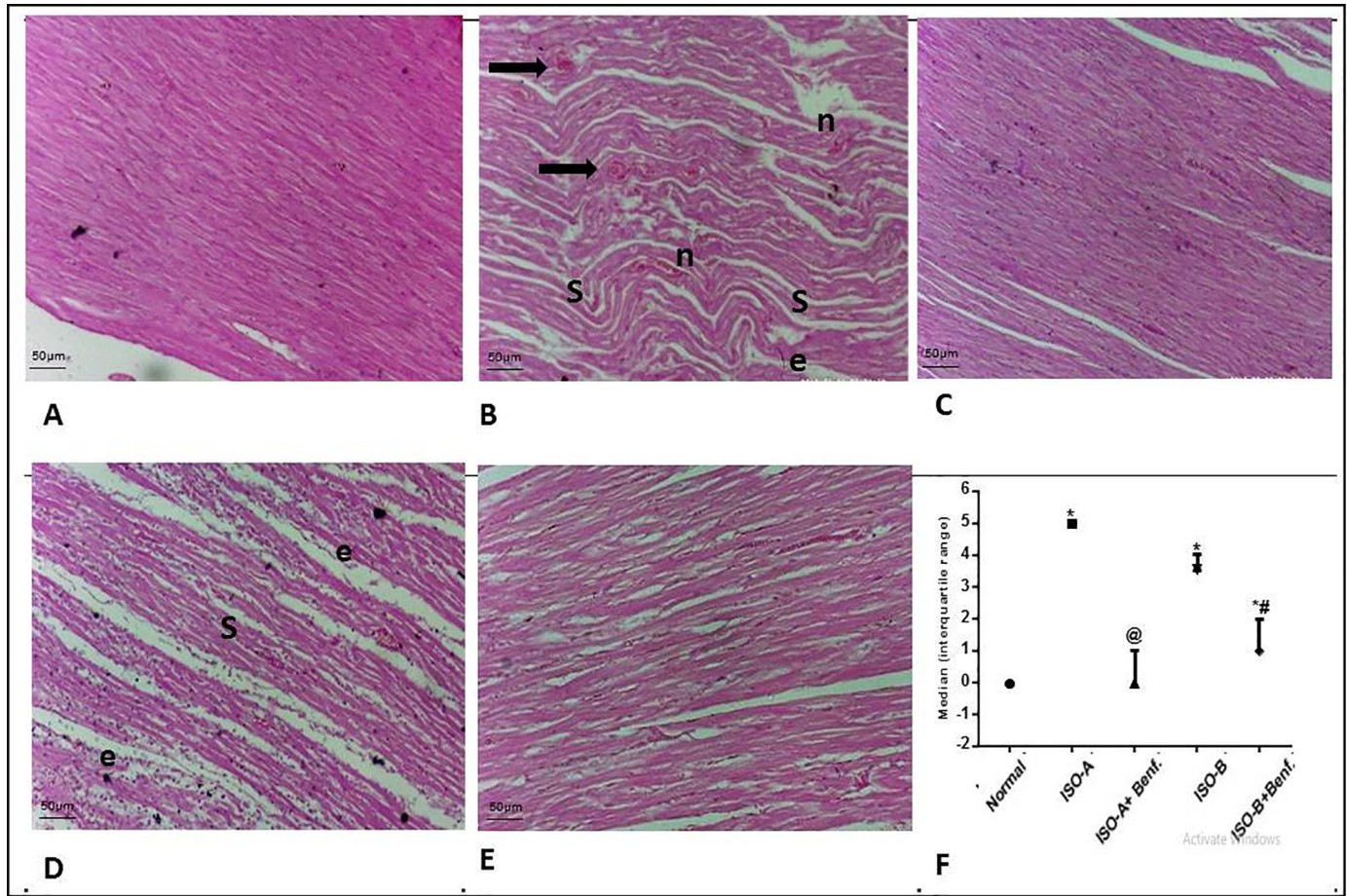

**Fig 5. Effect of benfotiamine pre- and post-treatments on histological damage in ISO-induced MI in rats.** (A–F) Cardiac specimens were stained with H&E (magnification x200). Normal group (A) showed normal histological structure of cardiac muscle. ISO-A treatment group (B) showed severe cardiac insult with atrophy, muscle shrinkage (s) edema (e) of cardiac muscle, marked inflammatory cells infiltration (n), extensive edema in-between muscle fibers (e), and marked number of apoptotic or degenerated myocytes filament (black arrows). Benfotiamine pre-treatment group (C) showed mostly normal cardiac muscle architecture. ISO-B treatment group (D) revealed shrinkage (s) of cardiac muscle with preserved nuclei all over sections examined, edema in between muscle fibers (e). Benfotiamine post-treatment (E) demonstrated almost normal cardiac muscle with only mild edema in-between muscle fibers. Myocardial score of damage expressed as median (interquartile changes) (F). Each value represents the median value [interquartile range] (n = 6). Statistical analysis was done using non-parametric One-Way ANOVA (Kruskal-Wallis test) followed by Dunn's multiple comparison test. * $p < 0.05$ vs. normal, @ $p < 0.05$ vs. ISO-A, # $p < 0.05$ vs. ISO-B. ISO-A: ISO pre-treatment control; ISO-B: ISO post-treatment control; ISO-A+ Benf.: benfotiamine prophylactic; ISO-A+ Benf.: benfotiamine treatment.

## 4. Discussion

ISO has been reported as a well-standardized model for induction of myocardial injury which reveals similar metabolic and morphologic aberrations and ECG abnormalities comparable to those taking place in human MI. Thus, this model has been used extensively as an experimental model for the assessment of the cardioprotective effects of numerous drugs [19]. This model is also characterized by extraordinary technical simplicity, excellent reproducibility as well as acceptable mortality compared to other experimental models of MI which involve surgery as coronary artery ligation method [20]. The current study was designed to examine the beneficial effects of benfotiamine on myocardial damage when used as pre- and post-treatments in ISO-induced MI in rats. ISO-A and ISO-B groups involved in the current study displayed obvious inflammation, imbalance in oxidant/antioxidant markers and cardiac damage, which were confirmed by biochemical and histopathological examinations.

HWI recorded a significant increase in ISO groups, showing hypertrophy, which was significantly ameliorated by benfotiamine in pre- and post-treatments. The detected myocardial hypertrophy and oedema could be correlated to inflammatory cells invasion as well as increased fibrosis [21], as confirmed by histological examinations in the current study. Regarding ECG parameters, the ISO-A group revealed significant increases in HR, ST elevation and T wave inversion along with significant prolongation of QTc interval as well as QRS duration, indicating ischaemia and conduction abnormalities. Similarly, ISO-B group exerted significant changes in ECG parameters similar to ISO-A group, except for depression in ST segment amplitude, which could indicate the progression of ischaemia-induced MI. In context, a previous experimental study of ISO-induced myocardial tissue damage revealed several ECG abnormalities as a result of oxidative stress-induced cellular membrane damage and necrosis [21]. ST-segment elevation reflects the potential difference in the boundary between ischemic and non-ischemic zones and the consequent loss of cell membrane function. It was observed either in patient with acute myocardial ischemia or experimentally in ISO-induced MI [19]. On the other hand, pre- and post-treatments of benfotiamine succeeded in improving ECG abnormalities secondary to ISO administration. According to the literature, benfotiamine administration was found to be beneficial in attenuating diabetic cardiomyopathy, which was associated with reduced oxidative stress. Additionally, benfotiamine was previously demonstrated to improve diastolic dysfunction in diabetes-induced heart failure via activation of Akt/Pim-1-mediated survival pathway [12,22].

The effects of ISO-induced myocardial damage were also shown by significant increases in cTn-I and CK-MB levels in both ISO-A and ISO-B groups in comparison to the normal group. This elevation could be explained by the increase in cell membrane permeability, which allows cardiac enzymes to leak out and be detected in the bloodstream [10,23–25]. The results of the pre- and post-benfotiamine treatment groups revealed normalization of the cardiac enzyme markers; cTn-I and CK-MB. This effect could be attributed to the decline in ROS production with a consequent decrease in oxidative stress [12]. Similarly, the cardioprotective activity of benfotiamine was previously reported due to attenuation of cardiac cell death and enhancement of contractile function under hyperglycaemic conditions [26]

In the present study, there is an obvious positive association between the degree of oxidative stress and the severity of cardiac tissue damage. First, ISO-A and B groups showed a significant elevation of NADPH oxidase activity compared to the normal group which could be related to β-adrenoceptor stimulation provoking NADPH oxidase-derived ROS production in the heart [5]. Furthermore, NADPH oxidase has been reported to play a critically important role in ISO-induced ROS [10,27]. Benfotiamine pre- and post-treatments showed normalization and complete protection against the elevation of NADPH oxidase activity. Benfotiamine, a specific NADPH oxidase inhibitor, was found to not only inhibit the activity of NADPH oxidase but also reduce the expression of the enzyme and directly scavenge superoxide radical anions [28].

Second, ISO-A and ISO-B groups showed a significant increase in myocardial MDA content due to ROS generation as a result of the elevation of NADPH oxidase activity with subsequent depletion of cardiac GSH content in comparison to the normal group. The current results are in agreement with the work of other investigators [29,30]. Focusing on antioxidant markers, ISO-A group showed a non-significant change in GPx and SOD activities whereas the ISO-B group revealed a significant increase in the myocardial activity of GPx and SOD in comparison to normal rats. Activated lipid peroxidation is considered a crucial pathogenic event in MI [31] as well as it may clarify the association between increased production of MDA and the damaging effect noticed on myocardial cells as evidenced by the increased cardiac enzymes leakage. Moreover, the reduction of the antioxidant GSH content could be justified by their excessive utilization throughout the burst of ROS production [32]. The elevation

of the cardiac antioxidant activities of SOD and GPx after 14 days of ISO induction could justify the decline in MDA content in ISO-B group. This result is in agreement with a previous study stating an increase in the enzymatic antioxidant machinery after induction with ISO (1 mg/kg/day) for 10 days in rats, indicating that oxidative stress induced by ISO was able to upregulate the activity of the cardiac antioxidant enzymes such as SOD and GPx in order to facilitate rapid elimination of the accumulated ROS [33]. Thus, the increase in antioxidant enzyme levels after treatment with ISO could be explained by the adaptive response towards oxidative stress.

Regarding benfotiamine treatment in the pre- and post-treated groups, animals showed a significant reduction and normalization of tissue MDA content. Similarly, benfotiamine pre-treatment group revealed a significant increase in myocardial GSH content while benfotiamine post-treatment completely improved perturbations of GSH content. Benfotiamine was previously shown to improve the antioxidant capacity in addition to the reduction of lipid peroxidation in different experimental models [12,34–36]. Additionally, it was stated that benfotiamine reduced oxidative stress, activated eNOS and subsequently improved the integrity of vascular endothelium in experimentally-induced vascular endothelial dysfunction [37]. Benfotiamine reduces ROS formation by activation of the pentose phosphate pathway, which regenerates NADPH and is important for replenishing the major cellular antioxidants (GSH). This result was attributed to the action of benfotiamine as a direct antioxidant [38]. Therefore, in the present study, benfotiamine was found to protect the heart from the damaging effect of ISO and exert significant antioxidant activity via elevation of GSH and inhibition of MDA contents in both prophylactic and treatment groups compared to their ISO control groups.

While studying the benfotiamine effect, it was noticed that benfotiamine post-treatment completely improved perturbation of GPx and SOD activities with further significant increase in their values in comparison to normal group. Likewise, the effect of benfotiamine on myocardial SOD activity showed a notable significant elevation in the pre-treatment group compared to the normal group. The potential antioxidant capacity as well as the free radical scavenging activity of benfotiamine was previously shown to be linked to the upregulation of cardiac GPx activity [11].

The results of the present study revealed a state of oxidative stress and inflammation, as both are considered as the main causative factors implicated in ISO-induced MI. In the present study, oxidative stress caused significant elevation of PKC and NF-$\kappa$B contents in ISO-treated groups. The latter is responsible for the upregulation of inflammatory cytokines along with a significant elevation of MMP-9 compared to the normal group. These data are in agreement with earlier reports [39–41]. β-adrenoceptor stimulation by ISO disperses not only NADPH oxidase activity but also the PKC and MMP-9 pathways across the heart, which are mediated by ROS [6,42,43]. Thus, NADPH oxidase enzymes are specifically dedicated to the production of ROS that induce oxidative stress and inflammation [44].

In the current study, pre- and post-treatments with benfotiamine revealed marked protection and normalization of PKC and NF-$\kappa$B contents. Likewise, the benfotiamine pre-treatment normalized MMP-9 content, while the post-treated group showed only a significant reduction of its content compared to ISO-B group. These findings clearly demonstrated the anti-inflammatory properties of benfotiamine as reported earlier [11,45]. Benfotiamine was previously verified to inhibit not only NADPH oxidase activity but also the pathways of PKC and thus block the activation of NF-κB in patients with diabetes, hence preventing NF-κB dependent signalling events such as the transcription of inflammatory cytokines, chemokines, as well as other inflammatory markers [13,36,46]. Experimentally, benfotiamine prevented ocular inflammation induced by endotoxin in rats via supressing the NF-κB-mediated expression of inflammatory markers [13]. This evidence suggests that benfotiamine supplementation may

be used safely as a powerful antioxidant in different cardiovascular diseases involving oxidative stress and other inflammatory disorders.

The data of the present study revealed that administration of ISO resulted in upregulation of myocardial contents of caspase-8 and p53 in ISO-A and ISO-B groups compared to the normal group. The aforementioned results are in agreement with the work of other investigators [47–49]. Previous literature verified the initiation of apoptosis in the myocardium after ISO administration via caspase-8,9 and FAS genes [50,51]. This evidence could be related to the marked elevation of oxidative stress markers, depletion of endogenous antioxidants and stimulation of inflammatory as well as apoptotic pathways as a result of β-stimulation [21].

Our results indicated that pre- and post-treatment with benfotiamine effectively reduced the apoptosis and induced protection against damage produced by ISO administration, as shown by the reduced contents of caspase-8 and p53. Similarly, another study revealed that thiamine in addition to benfotiamine prevented apoptosis produced via hyperglycaemic conditions in retinal pericytes of humans [52]. Benfotiamine was previously shown to be a potent anti-inflammatory [28] and anti-apoptotic agent by augmentation of the antioxidant defence that further could reduce the genomic damage and enhance the expression of the anti-apoptotic proteins [11,37].

Regarding the histopathological examination, the cardiac damage score was in accordance with our ECG results and the biochemical changes observed in both ISO groups. The benfotiamine prophylactic group significantly improved the histopathological alterations with the cardiac muscle showing nearly normal structure. Similarly, the benfotiamine post-treatment group improved the histopathological picture compared to the ISO-B group, with a cardiac lesion score of 1. The superior results observed by benfotiamine pre-treatment could be explained most likely by the time element of 14 days of benfotiamine administration before induction of myocardial damage by ISO, which allowed better protection of heart muscle.

In conclusion, in the current investigation, ISO-induced cardiotoxicity was mediated via enhancement of NADPH oxidase as well as oxidative stress, inflammatory and apoptotic pathways, which eventually contributed to cardiac damage, remodelling and failure. Pre- and post-treatments with benfotiamine were effective in ameliorating ISO-induced MI by alleviating the state of oxidative stress, inflammation and apoptosis. Thus, benfotiamine, as an NADPH oxidase inhibitor, could be considered a promising agent for therapeutic and prophylactic interventions in different cardiovascular disorders, including ischaemia, myocardial tissue damage and heart failure. Finally, experimental and other clinical studies are warranted to verify the beneficial effectiveness of benfotiamine as a promising approach to ameliorate acute as well as chronic cardiovascular disorders.

## Acknowledgments

The authors are grateful to Dr. Rofanda Bekir, Department of Pathology, Faculty of Medicine, Helwan University, Egypt, for the kind help with histopathological examination and Dr. Marwan Abd-Elbaset, Assistant Researcher, Department of Pharmacology, Medical Research Division, National Research Center, Cairo, Egypt, for the kind assistance with performing ECG analysis.

## Author Contributions

**Conceptualization:** Lamiaa A. Ahmed, Omneya Galal, Dina F. Mansour, Aiman El-Khatib.

**Data curation:** Lamiaa A. Ahmed, Omnia F. Hassan.

**Formal analysis:** Lamiaa A. Ahmed, Omnia F. Hassan, Omneya Galal, Dina F. Mansour.

**Funding acquisition:** Omnia F. Hassan.

**Investigation:** Omnia F. Hassan, Dina F. Mansour.

**Methodology:** Lamiaa A. Ahmed, Omnia F. Hassan, Omneya Galal, Dina F. Mansour.

**Project administration:** Lamiaa A. Ahmed, Aiman El-Khatib.

**Resources:** Lamiaa A. Ahmed, Omnia F. Hassan, Aiman El-Khatib.

**Software:** Omnia F. Hassan.

**Supervision:** Lamiaa A. Ahmed, Omneya Galal, Dina F. Mansour, Aiman El-Khatib.

**Validation:** Omnia F. Hassan, Dina F. Mansour.

**Visualization:** Omnia F. Hassan.

**Writing – original draft:** Omnia F. Hassan.

**Writing – review & editing:** Lamiaa A. Ahmed, Omneya Galal, Dina F. Mansour, Aiman El-Khatib.

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
