## [Decision Letter · Decision Letter 0]

13 Jan 2020

PONE-D-19-33705

Beneficial effects of benfotiamine, a NADPH oxidase inhibitor, in isoproterenol-induced myocardial infarction in rats

PLOS ONE

Dear Dr Ahmed,

Thank you for submitting your manuscript to PLOS ONE. After careful consideration, we feel that it has merit but does not fully meet PLOS ONE’s publication criteria as it currently stands. Therefore, we invite you to submit a revised version of the manuscript that addresses the points raised during the review process. Please note, that lack of novelty (reviewer 2) is not a valid criticism in PLOS ONE.

We would appreciate receiving your revised manuscript by Feb 27 2020 11:59PM. To enhance the reproducibility of your results, we recommend that if applicable you deposit your laboratory protocols in protocols.io, where a protocol can be assigned its own identifier (DOI) such that it can be cited independently in the future. For instructions see: http://journals.plos.org/plosone/s/submission-guidelines#loc-laboratory-protocols

We look forward to receiving your revised manuscript.

Kind regards,

Michael Bader

Academic Editor

PLOS ONE

Journal Requirements:

2. At this time, we ask that you please provide the product number and any lot numbers provided with the Isoproterenol-HCl and benfotiamine purchased from Sigma-Aldrich in the study."

3. At this time, we request that you  please report additional details in your Methods section regarding animal care, as per our editorial guidelines:

(a) Please include the method of euthanasia  

(b) Please describe the care received by the animals, including the frequency of monitoring and the criteria used to assess animal health and well-being.

Thank you for your attention to these requests.

4. We ask that you please provide scale bars on the microscopy images presented in Figure 5 and refer to the scale bar in the corresponding Figure legend.

Reviewers' comments:

Reviewer's Responses to Questions

**Comments to the Author**

1. Is the manuscript technically sound, and do the data support the conclusions?

Reviewer #1: Yes

Reviewer #2: Partly

2. Has the statistical analysis been performed appropriately and rigorously? 

Reviewer #1: Yes

Reviewer #2: I Don't Know

3. Have the authors made all data underlying the findings in their manuscript fully available?

Reviewer #1: Yes

Reviewer #2: Yes

4. Is the manuscript presented in an intelligible fashion and written in standard English?

Reviewer #1: Yes

Reviewer #2: No

5. Review Comments to the Author

Reviewer #1: Ahmed an coauthors elegantly analysed the effects of a NADPH oxidase inhibitor, benfotiamine, on isoproterenol-induced myocardial infarction in rats. They choose an acute therapeutic model and applied benfotiamine after induction of MI, or gave benfotiamine two weeks before induction of MI. Beside measurement of classic echocardiographic outcome parameters they analyzed several plasma markers by ELISA.

Observed effects were clear and are presented in a detailed fashion. There is no need for improvement of analyses.

Figures may be improved by more concise denomination of groups eg control, Iso-A, Iso-A+BT, Iso-B, Iso-B+BT instead of "prophylactic" or "treatment". For the measurement of parameters ELISA or colorimetric assays were used. Specify and sensitivity should be given for each test, as well as number of measurements per sample (N and n).

Detailed figure legends are missing and should be provided.

The discussion may be focussed to the main observed effects, as it appears cloudy and long. Katare et al published in 2010 comparable results by applying Benfotiamine in a streptocytozin mouse model. The reviewer would appreciate a detailed discussion of the 2010 results in comparison of the presented results. I feel these data should be considered as well. Kadare R et al. JMCC, 49, 625-538.

Reviewer #2: The current study explores the potential of benfotiamine (BFT) as cardio-protective agent via direct inhibition of NADPH oxidase (NOX). Although an interesting study, the authors fail to illustrate the novelty of this work. Previous reports also show a role for BFT as cardio-protective agent with (indirect) antioxidant properties, including inhibition of NOX activity and augmentation of antioxidant systems (e.g. GSH). The novelty of the current study is therfore not clear. There are also a number of issues that the authors should address:

1) A discussion of the method of ischemia induction used here, versus Langendorff or coronary artery ligation used in other studies, should be included - specifically how the methods would impact results.

2) As NOX is a central molecule in the study, the measurement of NOX enzyme content should be supplemented by activity assays. Determination of the NOX subunits and isoforms may also add value.

3) More detailed descriptions of the methods used is needed.

4) Why was PKC used as a marker of inflammation but not one of the "conventional" markers (e.g. IL-6 or TNF alpha)? Also, which PKC isoforms play a role?

5) Were all the data normally distributed?

6) Please indicate p-values when reporting significant differences in text.

7) A measurement of NADPH levels in the cardiac tissue may add an explanation for the results observed. This may facilitate a mechanistic explanation for the link between BFT and decreased NOX activity.

8) The manuscript may benefit from language editing as some grammatical errors occur throughout the text.

6. PLOS authors have the option to publish the peer review history of their article (what does this mean?). If published, this will include your full peer review and any attached files.

Reviewer #1: No

Reviewer #2: No

---

## [Author Response · Author response to Decision Letter 0]

20 Feb 2020

Ref: PONE-D-19-33705

Title: Beneficial effects of benfotiamine, a NADPH oxidase inhibitor, in isoproterenol-induced myocardial infarction in rats

Journal: PLOS ONE

We sincerely thank the Academic Editor and the respectable reviewers for their precious time and constructive criticisms. Their valuable comments were of great help in revising the manuscript and they have definitely improved it. 

Accordingly, the revised manuscript has been systematically enhanced with new information and additional interpretations. Following are the detailed responses to the reviewers' and Editor's comments:

Journal Requirements:

Response: All files’ names were revised

2) At this time, we ask that you please provide the product number and any lot numbers provided with the Isoproterenol-HCl and benfotiamine purchased from Sigma-Aldrich in the study."

Response: Done and highlighted

3) At this time, we request that you please report additional details in your Methods section regarding animal care, as per our editorial guidelines:

(a) Please include the method of euthanasia 

Response: This part was clarified and highlighted in the manuscript

(b) Please describe the care received by the animals, including the frequency of monitoring and the criteria used to assess animal health and well-being.

Response: This part was clarified and highlighted in the manuscript

4) We ask that you please provide scale bars on the microscopy images presented in Figure 5 and refer to the scale bar in the corresponding Figure legend.

 Response: This part was clarified and highlighted in the manuscript

5) Please include captions for your Supporting Information files at the end of your manuscript, and update any in-text citations to match accordingly. Please see our Supporting Information guidelines for more information.

No supporting information files are present in our submission

Review Comments to the Author

Reviewer one:

1. Figures may be improved by more concise denomination of groups eg control, Iso-A, Iso-A+BT, Iso-B, Iso-B+BT instead of "prophylactic" or "treatment".

We thank the reviewer for his/her important comment. These parts were modified in all figures

2. For the measurement of parameters ELISA or colorimetric assays were used. Specify and sensitivity should be given for each test. 

These parts were revised and highlighted in the manuscript

3. It is necessary to state the n (number of samples) used in each essay. 

Clarified and highlighted in the manuscript

4. Detailed figure legends are missing and should be provided.

Detailed figure legends were provided and highlighted at the end of manuscript after references section

5. The reviewer would appreciate a detailed discussion of the 2010 results in comparison of the presented results. I feel these data should be considered as well. Kadare R et al. JMCC, 49, 625-538.

This part was added and highlighted in the manuscript

Reviewer two:

1) A discussion of the method of ischemia induction used here, versus Langendorff or coronary artery ligation used in other studies, should be included - specifically how the methods would impact results.

The part was provided and highlighted in the manuscript

2) As NOX is a central molecule in the study, the measurement of NOX enzyme content should be supplemented by activity assays. Determination of the NOX subunits and isoforms may also add value.

NADPH oxidase activity is a relatively nonspecific activity displayed by several electron transfer enzymes and dehydrogenases as well. Therefore, assessing the NADPH oxidase activity of whole cell homogenates will not be accurate due to being heavily contaminated by all mitochondrial and cytosolicdehydrogenases(Laurindo, Fernandes & Santos 2008). Regarding NOX subunits, this would merit further investigation in a future study. 

3) More detailed descriptions of the methods used is needed.

Response: More detailed methodology for ECG and histopathology were provided and highlighted in the manuscript. ELISA kits used in the present study employed Double Antibody Sandwich Technique. This part was added and highlighted in the manuscript. On the other hand, detailed procedures provided by ELISA and colorimetric kits are usually not written as each differs from one company to another. Thus, it is usually written that procedures were done according to the manufacturer's instructions.

4) Why was PKC used as a marker of inflammation but not one of the "conventional" markers (e.g. IL-6 or TNF alpha)? Also, which PKC isoforms play a role?

PKC activation mediates inflammatory response signals through NF-κB pathway (Kim et al. 2013). The activation of NF-�B is responsible for the upregulation of inflammatory cytokines e.g. IL-6 or TNF alpha (Kany, Tilmann Vollrath & Relja 2019). Therefore, its elevation is an indicator of increase in the levels of inflammatory cytokines. Furthermore, ROS, produced through NADPH oxidase, activates NF-kB through various intermediates, including PKC (Panday et al. 2015). In the present study, total PKC was estimated which was previously shown to participate in cardiac remodeling, oxidative stress and arrhythmogenesis.

5) Were all the data normally distributed?

Response: Provided and added in the manuscript.

6) Please indicate p-values when reporting significant differences in text.

Response: Added and highlighted in the manuscript.

7) A measurement of NADPH levels in the cardiac tissue may add an explanation for the results observed. This may facilitate a mechanistic explanation for the link between BFT and decreased NOX activity.

NADPH is an important cofactor for NADPH oxidase. NADH and NADPH are very labile which would make their extraction very difficult for measurements by fluorescence and enzymatic methods(You et al. 2019).

8) The manuscript may benefit from language editing as some grammatical errors occur throughout the text.

Concerns the language editing, the manuscript has been subjected to peer-revision by Nature Research Editing Service and the certificate is attached with the submitted files. 

References: 

Kany, S., Tilmann Vollrath, J. & Relja, B. 2019, ‘Molecular Sciences Review Cytokines in Inflammatory Disease’, International Journal of Molecular Sciences, vol. 20, no. 6008, viewed 28 January 2020, www.mdpi.com/journal/ijms>.

Kim, H., Zamel, R., Bai, X.H. & Liu, M. 2013, ‘PKC Activation Induces Inflammatory Response and Cell Death in Human Bronchial Epithelial Cells’, PLoS ONE, vol. 8, no. 5.

Laurindo, F.R.M., Fernandes, D.C. & Santos, C.X.C. 2008, ‘Chapter 13 Assessment of Superoxide Production and NADPH Oxidase Activity by HPLC Analysis of Dihydroethidium Oxidation Products’, Methods in Enzymology, vol. 441, Academic Press Inc., pp. 237–60.

Panday, A., Sahoo, M.K., Osorio, D. & Batra, S. 2015, ‘REVIEW NADPH oxidases : an overview from structure to innate immunity-associated pathologies’, Cellular & Molecular Immunology, vol. 12, no. August 2014, pp. 5–23.

You, S.-H., Lim, H.-D., Cheong, D.-E., Kim, E.-S. & Kim, G.-J. 2019, ‘Rapid and sensitive detection of NADPH via mBFP-mediated enhancement of its fluorescence’, PLOS ONE, vol. 14, no. 2, viewed 28 January 2020, https://doi.org/10.1371/journal.pone.0212061.g001.

---

## [Decision Letter · Decision Letter 1]

3 Mar 2020

PONE-D-19-33705R1

Beneficial effects of benfotiamine, a NADPH oxidase inhibitor, in isoproterenol-induced myocardial infarction in rats

PLOS ONE

Dear Dr Ahmed,

Thank you for submitting your manuscript to PLOS ONE. After careful consideration, we feel that it has merit but does not fully meet PLOS ONE’s publication criteria as it currently stands. Therefore, we invite you to submit a revised version of the manuscript that addresses the points still raised by the reviewer 2.

We would appreciate receiving your revised manuscript by Apr 17 2020 11:59PM. To enhance the reproducibility of your results, we recommend that if applicable you deposit your laboratory protocols in protocols.io, where a protocol can be assigned its own identifier (DOI) such that it can be cited independently in the future. For instructions see: http://journals.plos.org/plosone/s/submission-guidelines#loc-laboratory-protocols

We look forward to receiving your revised manuscript.

Kind regards,

Michael Bader

Academic Editor

PLOS ONE

Reviewers' comments:

Reviewer's Responses to Questions

**Comments to the Author**

1. If the authors have adequately addressed your comments raised in a previous round of review and you feel that this manuscript is now acceptable for publication, you may indicate that here to bypass the “Comments to the Author” section, enter your conflict of interest statement in the “Confidential to Editor” section, and submit your "Accept" recommendation.

Reviewer #2: (No Response)

2. Is the manuscript technically sound, and do the data support the conclusions?

Reviewer #2: Partly

3. Has the statistical analysis been performed appropriately and rigorously? 

Reviewer #2: Yes

4. Have the authors made all data underlying the findings in their manuscript fully available?

Reviewer #2: Yes

5. Is the manuscript presented in an intelligible fashion and written in standard English?

Reviewer #2: Yes

6. Review Comments to the Author

Reviewer #2: Thank you to the authors for responding to the comments from the first review. The manuscript is strengthened by the additions. However, some major concerns still persist and these points have to be addressed before the manuscript is of a high enough standard to be considered for publication. The authors should address the following points and clearly indicate in their response where in the manuscript these changes were made (line and page):

1. Please indicate in the discussion section how the simulated MI with ISO treatment differs from methods where ischemia is simulated - similarities or differences in results and parameters measured here (ECG and blood markers of cardiac damage vs. more direct measures of ischemic damage, e.g. infarct size). Also indicate how benfotiamine offers cardio-protection, making reference to previous studies (PMID: 20107192; 24286628).

2. As the authors strongly conclude that NOX activation is the primary oxidative stress-inducing mechanism, some measure of its activity is warranted. The concerns of non-specific measures may be addressed by including specific NOX and other system inhibitors in the experimental setup. Also, the premise of the article, as mentioned in line 343 as well as the title, is that benfotiamine exerts protective effects as a "NADPH oxidase inhibitor". The inhibitory effect is, however, indirect via transketolse activation. This distinction should be made clearer and all the more reason to provide stronger experimental evidence for NOX activation. Please also discuss results in context with previous studies (e.g. PMID 24747137).

7. PLOS authors have the option to publish the peer review history of their article (what does this mean?). If published, this will include your full peer review and any attached files.

Reviewer #2: No

---

## [Author Response · Author response to Decision Letter 1]

14 Apr 2020

Ref: PONE-D-19-33705

Title: Beneficial effects of benfotiamine, a NADPH oxidase inhibitor, in isoproterenol-induced myocardial infarction in rats

Journal: PLOS ONE

We sincerely thank the respectable reviewer for his precious time and constructive criticisms. His valuable comments were of great help in revising the manuscript and definitely improved it. 

Accordingly, the revised manuscript has been systematically enhanced with new information and additional interpretations. The followings are the detailed responses to the reviewer's comments:

Reviewer's Comments to the Author

Reviewer two:

1) Please indicate in the discussion section how the simulated MI with

ISO treatment differs from methods where ischemia is simulated -

similarities or differences in results and parameters measured here

(ECG and blood markers of cardiac damage vs. more direct measures of

ischemic damage, e.g. infarct size). Also indicate how benfotiamine

offers cardio-protection, making reference to previous studies (PMID:

20107192; 24286628).

These parts was provided in the discussion section as recommended by the reviewer. Suggested reference was added. The written parts were highlighted in the manuscript

2) As the authors strongly conclude that NOX activation is the primary

oxidative stress-inducing mechanism, some measure of its activity is

warranted. The concerns of non-specific measures may be addressed by

including specific NOX and other system inhibitors in the experimental

setup. Also, the premise of the article, as mentioned in line 343 as

well as the title, is that benfotiamine exerts protective effects as a

"NADPH oxidase inhibitor". The inhibitory effect is, however, indirect

via transketolse activation. This distinction should be made clearer

and all the more reason to provide stronger experimental evidence for

NOX activation. Please also discuss results in context with previous

studies (e.g. PMID 24747137). 

As recommended by the reviewer, NADPH oxidase activity was measured by using the lucigenin chemiluminescence method (Matsui et al. 2006) and this part was highlighted in the manuscript. The direct and indirect inhibition of NADPH oxidase by benfotiamine were also clarified and highlighted in the introduction.

References

Matsui, H., Shimosawa, T., Uetake, Y., Wang, H., Ogura, S., Kaneko, T., Liu, J., Ando, K. & Fujita, T. 2006, ‘Protective effect of potassium against the hypertensive cardiac dysfunction: Association with reactive oxygen species reduction’, Hypertension, vol. 48, no. 2, pp. 225–31, viewed 19 March 2020, http://www.ncbi.nlm.nih.gov/pubmed/16818802.

---

## [Editor Report · Decision Letter 2]

15 Apr 2020

Beneficial effects of benfotiamine, a NADPH oxidase inhibitor, in isoproterenol-induced myocardial infarction in rats

PONE-D-19-33705R2

Dear Dr. Ahmed,

We are pleased to inform you that your manuscript has been judged scientifically suitable for publication and will be formally accepted for publication once it complies with all outstanding technical requirements.

With kind regards,

Michael Bader

Academic Editor

PLOS ONE
---

## [Editor Report · Acceptance letter]

22 Apr 2020

PONE-D-19-33705R2 

Beneficial effects of benfotiamine, a NADPH oxidase inhibitor, in isoproterenol-induced myocardial infarction in rats 

Dear Dr. Ahmed:

I am pleased to inform you that your manuscript has been deemed suitable for publication in PLOS ONE. Congratulations! Your manuscript is now with our production department. 

With kind regards,

on behalf of

Prof. Michael Bader 

Academic Editor

PLOS ONE